# TLR9/MyD88/NF-κB signaling mediates mental stress-induced exacerbation of psoriasis through immune dysregulation in a mouse model

Qiuhe Song[1ʘ], Dongyang Li[2ʘ], Zhihao Yuan[1,3], Chaowen Zhang[1], Jianqiao Wang[1], Fangfang Liao[1], Pengfei Xu[1], Qipeng Xiao[1]*

1 Department of Dermatology, Affiliated Hospital of Jiujiang University, Jiujiang, Jiangxi, China,
2 Department of Pain Medicine, Affiliated Hospital of Jiujiang University, Jiujiang, Jiangxi, China,
3 Medical School of Nanchang University, Nanchang, Jiangxi, China

ʘ The authors contributed equally to this work.
* 15070907521@163.com

## Abstract

### Objective

Psoriasis is a chronic inflammatory autoimmune disease that affects physical and mental health. Mental stress has been shown to exacerbate human psoriasis by unknown mechanism.

### Methods

Peripheral blood mononuclear cells (PBMCs) were collected from patients with psoriasis and mental stress-treated psoriatic mice. The expression levels of TLR9/MyD88/NF-κB pathway-related molecules were analyzed by qRT-PCR and western blotting. Histological examination of skin lesions was examined using hematoxylin-eosin staining. The ratios of Treg/CD4⁺T cells and Th17/Treg cells were determined by flow cytometry. The associations among mental stress, the TLR9/MyD88/NF-κB pathway, and psoriasis were explored using pharmacological inhibitors and lentiviral transfection.

### Results

Our findings demonstrated a significant upregulation of TLR9/MyD88/NF-κB pathway-associated molecules in the PBMCs of psoriasis patients, accompanied by elevated expression of inflammatory factors. These observations were validated using a mouse model of psoriasis. Notably, mental stress was shown to activate the TLR9/MyD88/NF-κB pathway and enhance inflammatory factor production, while simultaneously increasing the Th17/Treg ratio and decreasing the Treg/CD4⁺T ratio. Therapeutic interventions including antipsychotic sertraline, pathway-specific inhibitors, and lentiviral transfection significantly ameliorated inflammatory markers and improved psoriasis severity grading.

**Data availability statement:** The datasets generated and/or analyzed during the current study are available in Zenodo, the open-access repository operated by CERN, [https://zenodo.org/records/18069564?token=eyJhbGciOiJIUzUxMiJ9.eyJpZCI6ImZjYzdkNmNlLTFhZjktNDE3ZS05YzdiLTI2NjhmM2U4MWY1OSIsImRhdGEiOnt9LCJyYW5kb20iOiJmYTBjMmIyZDAyNjk2Nzg0ODRkNmNlZDRhMTExNDFkZCJ9.HeVpzwu5n4OdiSw3toFMjTZBCfdZc_uQMNAjmFJ4yxSigNemmWOl5bSr0zyMPHrk74bYpXBmdUajlgSLbbN7-w]. The DOI is: https://doi.org/10.5281/zenodo.18069564.

**Funding:** This study was supported by National Natural Science Foundation of China (No. 82260283) and The Science and Technology Plan Project of Provincial Health Commission in 2025 (No. 202510823). The funders had no role in study design, data collection and analysis, decision to publish, or preparation of the manuscript.

**Competing interests:** The authors have declared that no competing interests exist.

## Conclusion

The results of this study demonstrates that mental stress induces inflammation and immune dysregulation, exacerbating psoriasis progression. These findings provide valuable insights into the pathophysiological mechanisms underlying psoriasis progression, particularly the mental stress-mediated immunoregulatory axis.

---

## Introduction

Psoriasis is a chronic immune-mediated skin disorder, characterized by cutaneous and/or articular manifestations. The most common form of psoriasis is psoriasis vulgaris, which results from genetic susceptibility. This complex condition presents considerable clinical challenges due to its high prevalence and frequent association with various systemic comorbidities [1–3]. It is also associated with several important diseases, including depression. Immunological and genetic studies have identified interleukin 17 (IL-17) and IL-23 as key drivers of the pathogenesis of psoriasis [4]. Risk factors for psoriasis can be classified into two categories, namely extrinsic and intrinsic risk factors [2]. Extrinsic factors include mechanical stress [5,6], exposure to air pollutants and the Sun [7,8], drug abuses [9], infections [10,11], and lifestyle. Intrinsic factors include obesity [12], diabetes [13], dyslipidemia [14], hypertension [15,16] and mental stress [17].

Mental stress plays a key role in the development of several skin diseases. The stigma associated with these conditions may exacerbate psychological burden, thereby creating a vicious cycle that further contributes to the development of skin diseases [18]. The prevalence of psoriasis is notably higher among individuals who experience major stressful events [19]. Furthermore, mental stress is the second most important factor contributing to psoriasis exacerbation after seasonal changes [20]. Recent studies have suggested that mental stress modulates skin immune imbalance to promote psoriasis progression, which is associated with the Toll-like receptor (TLR) pathway [18,21], and regulates the expression of cytokines such as IL-17 and IL-23 [22–26]. Th17 cells exacerbate skin inflammatory infiltration by secreting pro-inflammatory cytokines such as IL-17 and IL-23, whereas regulatory T (Treg) cells maintain immune homeostasis through the secretion of anti-inflammatory cytokines including IL-10 and transforming growth factor-β (TGF-β). The dynamic balance between these two cell subsets, reflected by the Th17/Treg ratio, directly regulates the inflammatory severity and disease progression of psoriasis [27]. Clinical studies have confirmed that the Th17/Treg ratio is significantly elevated in the peripheral blood and lesional tissues of psoriasis patients [28]. Therefore, this ratio is regarded as a core indicator for evaluating immune dysregulation in psoriasis. Meanwhile, as a critical subset of CD4$^+$T cells, Treg/CD4$^+$T ratio directly reflects the integrity of immune regulatory function. Under the pathological conditions of psoriasis, the proliferation, activation, and immunosuppressive capacity of Treg cells are impaired, leading to a decreased Treg/CD4$^+$T ratio. This reduction further weakens the inhibitory effect on aberrant inflammatory responses [29]. TLRs are extensively expressed

on immune cells and cellular membranes [30]. These receptors are activated through distinct signaling pathways, primarily mediated by either myeloid differentiation primary response 88 (MyD88) or TIR-domain-containing adaptor-inducing interferon-beta. These signaling cascades ultimately lead to enhanced production and secretion of inflammatory cytokines, thereby modulating the immune response and maintaining organismal homeostasis [31]. Furthermore, NF-κB plays an important role as a downstream signal of TLRs [32,33].

Emerging evidence has demonstrated that mental stress modulates the expression of local TLRs in the cerebral cortex, triggering disruptions in regional immune homeostasis through alterations in cytokine profiles and immune cell population dynamics [34]. These neuroimmune changes are closely associated with the development of psychiatric manifestations, including depression, anxiety, and suicidal behaviors [35,36]. As a crucial member of the TLR family, the abnormal expression of TLR9 is observed in various stress-related disorders. For example, Hung et al.[37] identified elevated levels of TLR9 in peripheral blood mononuclear cells (PBMCs) isolated from patients with major depressive disorder. Furthermore, Zhang et al.[38] provided experimental evidence indicating that exposure to chronic stress leads to an upregulation of TLR9 expression in murine blood samples, accompanied by an increase in TGF-β production. Collectively, these findings suggest that TLR9 serves as a crucial mediator of mental stress-induced neuroimmune dysregulation, and its altered expression patterns can significantly influence downstream cytokine signaling pathways within the nervous system. Notably, as a crucial hub bridging innate and adaptive immunity, the TLR9/MyD88/NF-κB signaling pathway modulates the Th17/Treg balance through multiple mechanisms. On one hand, TLR9/MyD88-mediated NF-κB pathway activation can promote dendritic cells to secrete cytokines such as IL-23 and IL-1β, which specifically induce the differentiation of naive CD4$^+$T cells into Th17 cells [39]. On the other hand, sustained NF-κB activation suppresses the expression of the Foxp3 transcription factor in Treg cells, thereby impairing the stability and immunosuppressive function of Treg cells [40]. Accumulating evidence has verified that in autoimmune disease models, inhibitors targeting the TLR9/NF-κB pathway can remarkably reduce the Th17/Treg ratio while elevating the Treg/CD4$^+$T ratio, ultimately alleviating inflammatory responses [41]. However, the interaction between mental stress and TLR9/MyD88/NF-κB signaling pathway as well as the possible regulatory mechanism during the progression of psoriasis are still rarely reported.

In this study, an imiquimod-induced psoriasis mouse model of mental stress was established. We conducted a preliminary investigation of the effects of mental stress on immune dysregulation and explored the underlying mechanism. The results of this study may provide valuable insights into the role of mental stress on the development of psoriasis, which help alleviate psychological distress and improve the clinical symptoms associated with psoriasis.

## Materials and methods

### Patients and PBMCs isolation

A total of eight patients diagnosed with psoriasis were enrolled in our hospital. Concurrently, eight healthy individuals matched for age and sex served as the control group. Peripheral blood samples from all participants were collected. Within four hours after blood collection, PBMCs were isolated from a 10 mL heparinized sample using a human peripheral blood monocyte isolation solution kit (Solarbio, Beijing, China). All participants were provided with detailed information about the study and had agreed by giving their written consent prior to involvement. Ethical approval for this study was granted by the Ethics Committee of Affiliated Hospital of Jiujiang University (No. JJU20220038), and the research was conducted in accordance with the principles set forth in the Declaration of Helsinki. The recruitment period started on 01/01/2023 and ended on 31/12/2024.

### Animal model and treatment

The animal studies were carried out following the principles set forth in the NIH Guide for the Care and Use of Laboratory Animals and received approval from the Ethics Committee of Affiliated Hospital of Jiujiang University (No. JJU20220038).

From SLAC Laboratory Animal (Shanghai China), SPF grade BALB/C male mice (weighing 18-22g) were purchased. Following acclimation, the mice were randomly assigned into 10 groups (n = 6): the control, model, model + mental stress (mental stress), model + mental stress + Sertraline (Sertraline), model + mental stress + ODN2088 (ODN2088), model + mental stress + ST2825 (ST2825), model + mental stress + PDTC (PDTC), model + mental stress + LV-shNC (LV-shNC), model + mental stress + LV-shTLR9 (LV-shTLR9) and model + mental stress + LV-shMyD88 (LV-shMyD88) groups.

For the induction of mental stress, the internationally recognized method of solitary stress was employed in this study with slight modifications [42,43]. Detailed procedures were as follows: specialized solitary cages (12 cm × 10 cm × 15 cm) were used, which provided only enough space for mice to move comfortably (i.e., allow free turning but restrict excessive locomotion) to induce mild physical constraint without causing physical injury. Each solitary cage was placed inside a larger transparent cage (40 cm × 30 cm × 25 cm), and the cage was covered with a light-impermeable black hood to create a dark environment, minimizing visual stimulation and simulating an isolated, unfamiliar context. No food or water was placed in the solitary cages during the stress period to avoid confounding effects of nutrient intake on stress responses. The solitary stress was initiated one week after mice adaptation to the animal facility and continued for 10 consecutive weeks. To avoid habituation, stress exposure was conducted from 7:00 to 9:00 a.m. three times per week (Monday, Wednesday, and Friday), with each session lasting 2 h. Immediately after each 2-hour stress session, mice were returned to their original colony cages (30 cm × 20 cm × 15 cm) with *ad libitum* access to food and water. The colony cages were maintained in a standardized environment (22 ± 1°C, 50 ± 5% relative humidity, 12-h light/dark cycle), ensuring consistent environmental cues except for the stress intervention. Psoriasis lesions were induced using imiquimod during the final week of the 10-week stress period [25,26]. The imiquimod-induced psoriasis mouse model was established as follows: mice were maintained on a regular diet, and psoriasis-like skin lesions were induced by imiquimod cream (Sigma-Aldrich, St. Louis, MO, USA) for a duration of 7 consecutive days. The control group received vaseline treatment. Additionally, drug administration was conducted as follows: Sertraline (antidepressant, 10 mg/kg; MedChemExpress, Monmouth Junction, NJ, USA) was administered via daily intraperitoneal injection at a volume of 200 µL per mouse; ODN2088 (TLR9 inhibitor; MedChemExpress) treatment involved a daily intraperitoneal injection of 200 µL (1 mg/mL); ST2825 (MyD88 inhibitor; MedChemExpress) treatment consisted of a daily tail vein injection of 200 µL (1 mg/mL); PDTC ((NF-κB p65 inhibitor; MedChemExpress) treatment included a daily intraperitoneal injection of 200 µL (10 mg/mL). Additionally, lentivirus short hairpin RNA TLR9 (LV-shTLR9), LV-shMyD88, alongside a shRNA control (LV-NC) ($4 \times 10^8$ transfection units; 50 µL; Weizhen Biosciences, Jinan, China) were injected into the tail vein. For the mice in the model group, an equivalent volume of normal saline was administered. The entire procedure lasted for a continuous period of 7 days. Throughout the experimental period, mice were monitored daily for general health status. Psoriasis area and severity index (PASI) score was identified as previously described [44]. Sucrose preference test and tail suspension test were used asses the depressive and anxiety-like behaviors of mice in each group. All behavioral tests were video-recorded and analyzed by two independent observers in a blinded manner.

At the conclusion of the experiment, animals were euthanized with overdose of pentobarbital sodium (200 mg/kg). Prior to euthanasia, the animals were anesthetized with isoflurane (5% for induction and 2–3% for maintenance) to minimize stress and discomfort. Whole blood samples from eyeball (200 µL) and skin tissues in lesions were collected for succeeding tests. With the aid of mouse peripheral blood monocyte isolation solution kit (Solarbio, Beijing, China), PBMCs from mice were isolated. Throughout the experiment, efforts were made to minimize animal suffering, including proper housing conditions, handling, and monitoring of animal well-being.

## Hematoxylin and eosin (HE) staining

Skin tissues were fixed, embedded, and sectioned. After deparaffinization with xylene (twice for 10 min each) and gradient ethanol hydration, sections were stained with hematoxylin (Beyotime, Shanghai, China) for 5 min, rinsed with alkaline

PBS, and then stained with eosin (Beyotime) for 5 min at room temperature. Following brief rinsing with distilled water, sections were dehydrated through gradient ethanol and washed twice in xylene. Finally, the sections were sealed using a neutral adhesive, and subsequently photographed under a microscope (Olympus, Tokyo, Japan) and meticulously recorded.

## Flow cytometric analysis

Flow cytometric analysis was employed to determine the frequencies of Th17 cells, Treg cells and CD4$^+$T cells. PBMCs were incubated with FITC-labeled anti-mouse CD4 (BioLegend, San Diego, CA, USA) for 15 min at 4°C for CD4$^+$T detection. To identify Treg frequency, FITC-labeled anti-mouse CD4, APC-labeled anti-mouse CD25 (BioLegend) and Alexa Fluor®700-conjugated anti-mouse Foxp3 (BioLegend) were incubated with PBMCs for 15 min at 4°C. To evaluate the frequency of Th17 cells, we employed FITC-labeled anti-mouse CD4 antibody to incubate with PBMCs, followed by incubation with PE-conjugated anti-IL-17A (Invitrogen, Carlsbad, CA, USA) for 15 min at 4°C. Following staining, the cells were washed and resuspended in a buffer containing 1% formaldehyde. All stained cells were analyzed using a flow cytometer (BD Bioscience, San Jose, CA, USA).

## qRT-PCR

After extracting total RNA from skin tissues or PBMCs with TRIzol reagent (Yeasen Biotechnology, Shanghai, China), the purity and concentration of RNA were determined. Following this step, the RNA was reverse transcribed into cDNA with Hifair® II 1st Strand cDNA Synthesis SuperMix (Yeasen Biotechnology). Subsequently, the synthesized cDNA, the corresponding primers and DEPC water were added in a 20 μL system, followed by conducting PCR analysis using a Hieff® qPCR SYBR Green Master Mix (Yeasen Biotechnology). The relative expression level of the target gene was determined by the 2$^{-\Delta\Delta Ct}$ method, which is normalized to GAPDH. The primers of genes were shown in **Table 1** and **Table 2**.

## Western blotting analysis

PBMCs were added into RIPA lysis buffer (Beyotime) to extract total protein. Quantification and sample preparation were performed after complete protein extraction. After loading, the proteins were separated by SDS-PAGE, transferred to the PVDF membrane, and blocked with 5% skimmed milk (Beyotime). Next, the membrane was incubated overnight at 4°C

**Table 1. The primers of target genes (Human).**

| Name | Sequence（5'-3'） |
|---|---|
| GAPDH | TCAAGAAGGTGGTGAAGCAGG |
| | TCAAAGGTGGAGGAGTGGGT |
| MyD88 | ACCCAGCATTGGTGCCG |
| | GGTTGGTGTAGTCGCAGACA |
| TLR9 | TCCTGCCCAAACTGGAAGTC |
| | GCTAAGGTTGAGCTCTCGCA |
| NF-κB p65 | TTTTCGACTACGCGGTGACA |
| | GTTACCCAAGCGGTCCAGAA |
| TGF-β | GGACCAGTGGGGAACACTAC |
| | TAAAGCAGGTTCCTGGTGGG |
| IL-10 | AAGACCCAGACATCAAGGCG |
| | AGGCATTCTTCACCTGCTCC |
| IL-17 | TAATGGCCCTGAGGAATGGC |
| | AGGAAGCCTGAGTCTAGGGG |

**Table 2. The primers of target genes (mouse).**

| Name | Sequence（5'-3'） |
|---|---|
| GAPDH | AAGAGGGATGCTGCCCTTAC |
| | ACGGCCAAATCCGTTCACA |
| TLR9 | CTCCAACCGTATCCACCACC |
| | GAGAAGTGCAGGGGGCTAAG |
| MyD88 | CCGCCTATCGCTGTTCTTGA |
| | GCCAGGCATCCAACAAACTG |
| NF-κB p65 | ACTGGAGTTGTACGGCAGTG |
| | GGGGCTGATCCCGTTGATTT |
| TGF-β | ACTGGAGTTGTACGGCAGTG |
| | GGGGCTGATCCCGTTGATTT |
| IL-1β | TGCCACCTTTTGACAGTGATG |
| | AAGGTCCACGGGAAAGACAC |
| IL-6 | TCCGGAGAGGAGACTTCACA |
| | GTGACTCCAGCTTATCTCTTGGT |
| IL-17 | GCTGACCCCTAAGAAACCCC |
| | GAAGCAGTTTGGGACCCCTT |
| IL-23 | TGGAGCAACTTCACACCTCC |
| | GGCAGCTATGGCCAAAAAGG |
| IFN-γ | GAGGTCAACAACCCACAGGT |
| | GGGACAATCTCTTCCCCACC |

with the primary antibodies and then incubated with the corresponding secondary antibodies for 1h at room temperature. After the PVDF membrane was washed for 30 min, the proteins were developed with enhanced ECL reagent (Beyotime), and results were detected with a gel imaging system. The relative protein expression was calculated using the Image J system to read the grey values.

The dilution of primary antibody and secondary antibody was: TLR9 (1:1000; Cell Signaling, Boston, MA, USA), MyD88 (1:1000; Abcam, Cambridge, UK), NF-κB p65 (1:2000; Proteintech, Manchester, UK), p-NF-κB p65 (1:2000; Proteintech), TGF-β (1:1000; Cell Signaling), IL-1β (1:1000; Proteintech), IL-6 (1:1000; Abcam), IL-17 (1:1000; Proteintech), IL-23 (1:1000; Abcam), GAPDH (1:5000; Proteintech), goat anti-rabbit secondary antibody (1:10000; Abcam), and goat anti-mouse secondary antibody (1:10000; Abcam).

## Immunofluorescent test

Following dewaxing and rehydration, the skin tissue sections underwent antigen retrieval. This was succeeded by a blocking step with goat serum. The samples were then incubated with a primary antibody targeting p-NF-κB p65 (Proteintech), IL-1β (Proteintech), IL-6 (Abcam), IL-17 (Proteintech) and IL-23 (Abcam) (all at a dilution of 1:50). Afterward, they were conjugated with an Alexa Fluor 568 secondary antibody at a dilution of 1:200 (Invitrogen). Post DAPI staining, fluorescence microscopy images of the sections were captured using an Olympus fluorescence microscope.

## ELISA

According to the instructions of the corresponding commercial kits (Beyotime), the concentrations of TGF-β, IL-10, IL-17, IFN-γ, IL-1β, IL-6 and IL-23 in the serum of patients and/or mice were determined.

## Statistical analysis

All data were statistically analyzed using GraphPad Prism software (Graph Pad Software, Inc., San Diego, CA, USA) and expressed as mean ± standard deviation. The Student's t-test was used to compare two groups, while one-way analysis was used to compare multiple groups. The results were considered statistically significant when a $P$ value < 0.05.

## Results

### TLR9/MyD88/NF-κB signaling is activated in PBMCs of patients with psoriasis

As illustrated in Fig 1A-1C, we found that the mRNA expression of TLR9, MyD88, and NF-κB p65 was significantly increased in the PBMCs derived from patients with psoriasis than in those from healthy controls. Inflammatory cytokine levels in PBMCs and the serum of patients with psoriasis were also determined. We observed a significant increase in the levels of TGF-β and IL-17, whereas concurrently IL-10 levels were markedly decreased in PBMCs and serum of patients with psoriasis (Fig 1D-1I).

### Mental stress activates the TLR9/MyD88/NF-κB pathway, stimulates inflammation responses, and induces immune dysfunction in an imiquimod-induced psoriasis mouse model

To investigate the relationship between mental stress and the TLR9/MyD88/NF-κB pathway in psoriasis pathogenesis, we established an imiquimod-induced psoriasis mouse model, accompanied by mental stress. The images of the affected skin in the different groups were photographed and shown in **Fig 2A**. Sucrose preference test and tail suspension test respectively showed that there were no significant differences between control mice and psoriatic mice in sucrose preference rate and stationary time (**Fig 2B**). However, compared to the model group, mice in the mental stress group exhibited decreased sucrose preference rate and increased stationary time (**Fig 2B**). As illustrated in **Fig 2C**, black arrows indicate the epidermal layer, red arrows mark the dermal layer, and blue arrows identify the hair follicles. HE staining demonstrated that the control group displayed normal skin histology with a thin epidermal layer. In contrast, the model group exhibited distinct pathological characteristics, including marked epidermal thickening, hypertrophy of the stratum spinosum, downward protrusions of the epidermis, and pronounced infiltration of inflammatory cells within the dermal layer. Notably, these classic psoriatic pathological changes were markedly exacerbated following the mental stress intervention. Furthermore, we demonstrated that mental stress treatment can activate the expression of the TLR9/MyD88/NF-κB pathway, as evidenced by the increased expression of TLR9, MyD88, NF-κB and p-NF-κB in the mental stress group compared to the model group (Fig 2D-2G). Additionally, mental stress intervention also stimulated the expression of inflammation-related factors (TGF-β, INF-γ, IL-1β, IL-6, IL-17 and IL-23) (Fig 2H-2I). Flow cytometer was performed to analyze the populations of CD4$^+$T, Treg and Th17 cells, and then the ratios of Treg/CD4$^+$T and Th17/Treg were assessed. Taking **Fig 3A** as an example, the CD4$^+$T cell population was gated in the scatter plot (the area enclosed by the red box in the plot). Within the pre-gated CD4$^+$T cell population, Treg cells were identified as the CD25$^+$FOXP3$^+$subset (located in the upper right [UR] quadrant). Therefore, Treg/CD4$^+$T ratio was calculated as (Treg cell population/CD4$^+$T cell population) × 100%, and the results are presented in the adjacent bar graph. We observed a significant decrease in the ratio of Treg/CD4$^+$T (**Fig 3A**) and a notable increase in the Th17/Treg ratio (**Fig 3B**) in psoriatic mice treated by mental stress. These results imply that under mental stress, the activation of the TLR9/MyD88/NF-κB pathway may be associated with inflammatory reactions and immune dysfunction during the progression of psoriasis.

### Treatment with the inhibitors of the TLR9/MyD88/NF-κB pathway attenuates inflammation and maintains immune balance in mental stress-treated psoriatic mice

To verify the potential role of mental stress in modulating immune homeostasis and inflammation via the TLR9/MyD88/NF-κB pathway, psoriatic mice were administered with the antidepressant Sertraline (positive control) and the specific

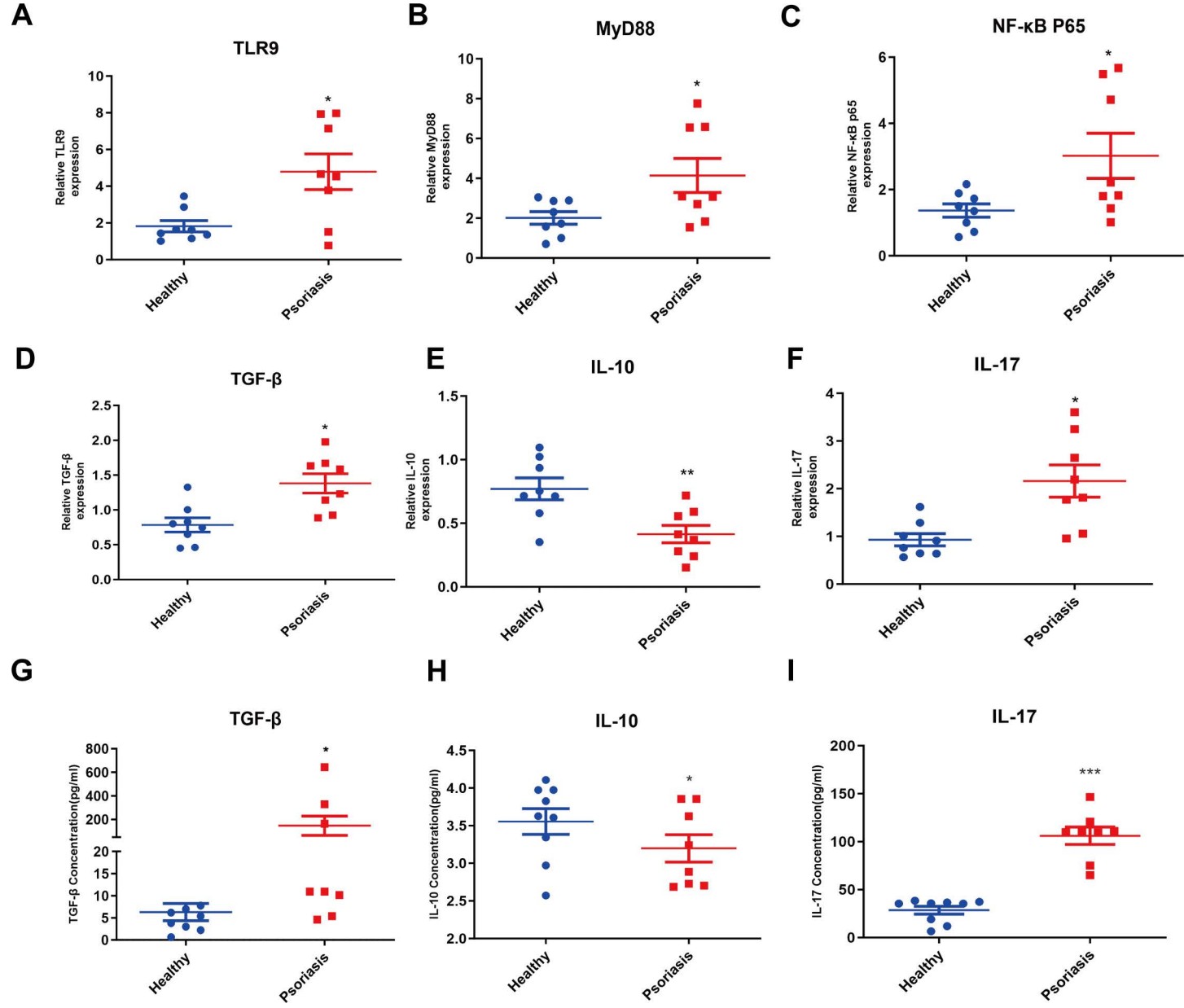

**Fig 1. TLR9/MyD88/NF-κB signaling is activated in PBMCs of patients with psoriasis.** (A-F): qRT-PCR was used to detect the expression level of TLR9, MyD88, NF-κB p65, TGF-β, IL-10 and IL-17 mRNA; (G-I): The concentrations of TGF-β, IL-10 and IL-17 were detected by ELISA. $^*P < 0.05$, $^{**}P < 0.01$, $^{***}P < 0.001$.

inhibitors of the TLR9/MyD88/NF-κB pathway. As illustrated in Fig 4A-B, we observed that the administration of Sertraline significantly inhibited the TLR9/MyD88/NF-κB pathway. Furthermore, ODN2088 markedly decreased TLR9 protein expression, whereas ST2825 and PDTC did not exhibit any significant effects. Treatment with both ODN2088 and ST2825 led to a reduction in MyD88 protein levels, whereas PDTC treatment had no notable influence. Notably, all three treatments—ODN2088, ST2825, and PDTC—resulted in a decrease in NF-κB and p-NF-κB protein levels. These findings suggest an underlying regulatory interaction within this signaling pathway. Moreover, we found that administration of Sertraline and

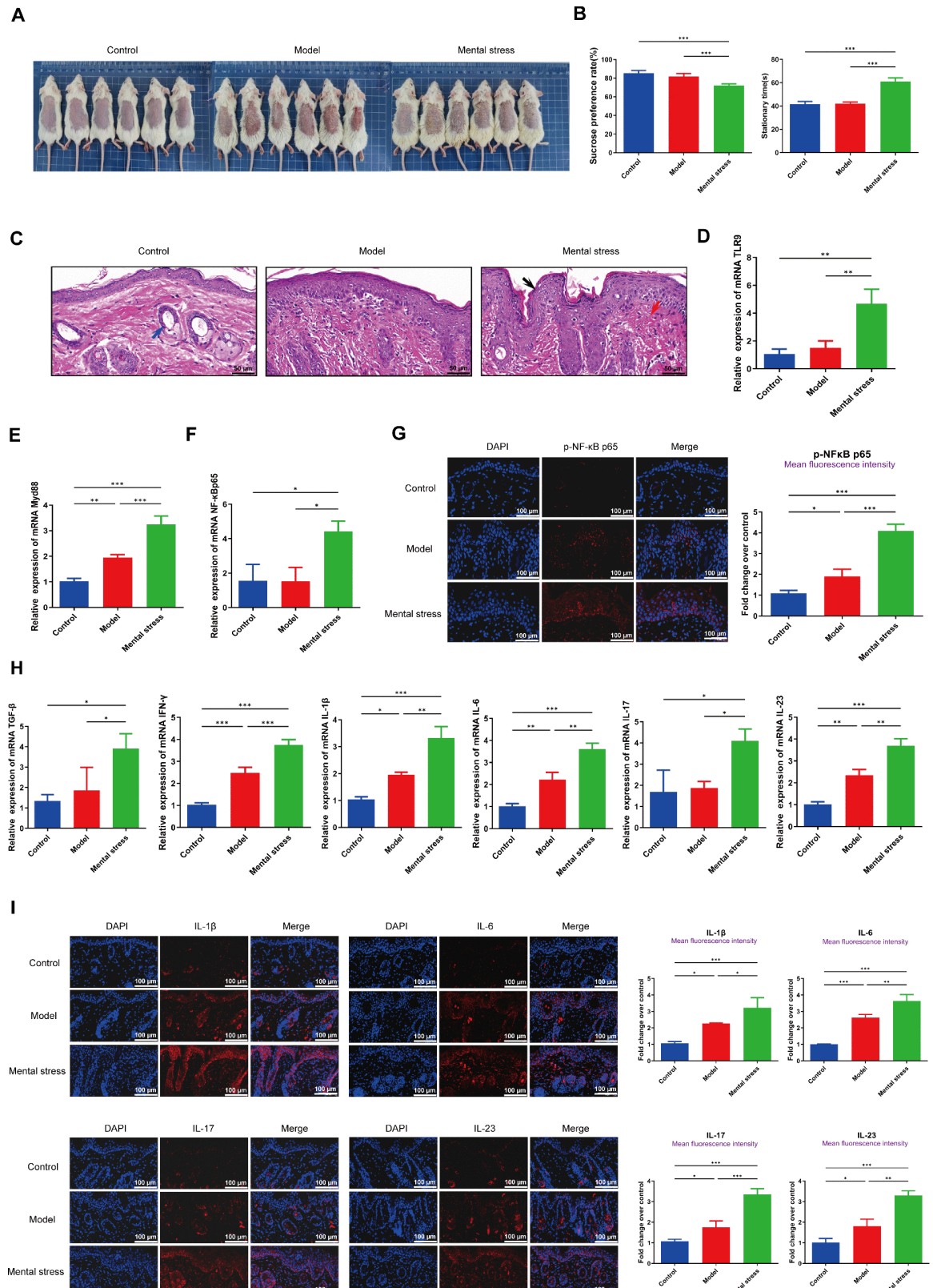

**Fig 2. Mental stress aggravates skin lesions and inflammation in psoriatic mice. (A)**: The images of the affected skin in the different groups were photographed. **(B)**: Sucrose preference rate and stationary time were assessed by sucrose preference test and tail suspension test respectively. **(C)**: HE

analysis of skin tissue in each group, scale bar = 50 μm; **(D-F)**: The expression levels of TLR9, MyD88, and NF-κB p65 in PBMCs of mice were detected by qRT-PCR; **(G)** The fluorescence intensity of p-NF-κB p65 was assessed by immunofluorescence test, scale bar = 100 μm; **(H)** The expression levels of TGF-β, IFN-γ, IL-1β, IL-6, IL-17 and IL-23 in PBMCs of mice were detected by qRT-PCR; **(I)** The fluorescence intensities of IL-1β, IL-6, IL-17 and IL-23 were assessed by immunofluorescence test, scale bar = 100 μm. $^*P < 0.05$, $^{**}P < 0.01$, $^{***}P < 0.001$.

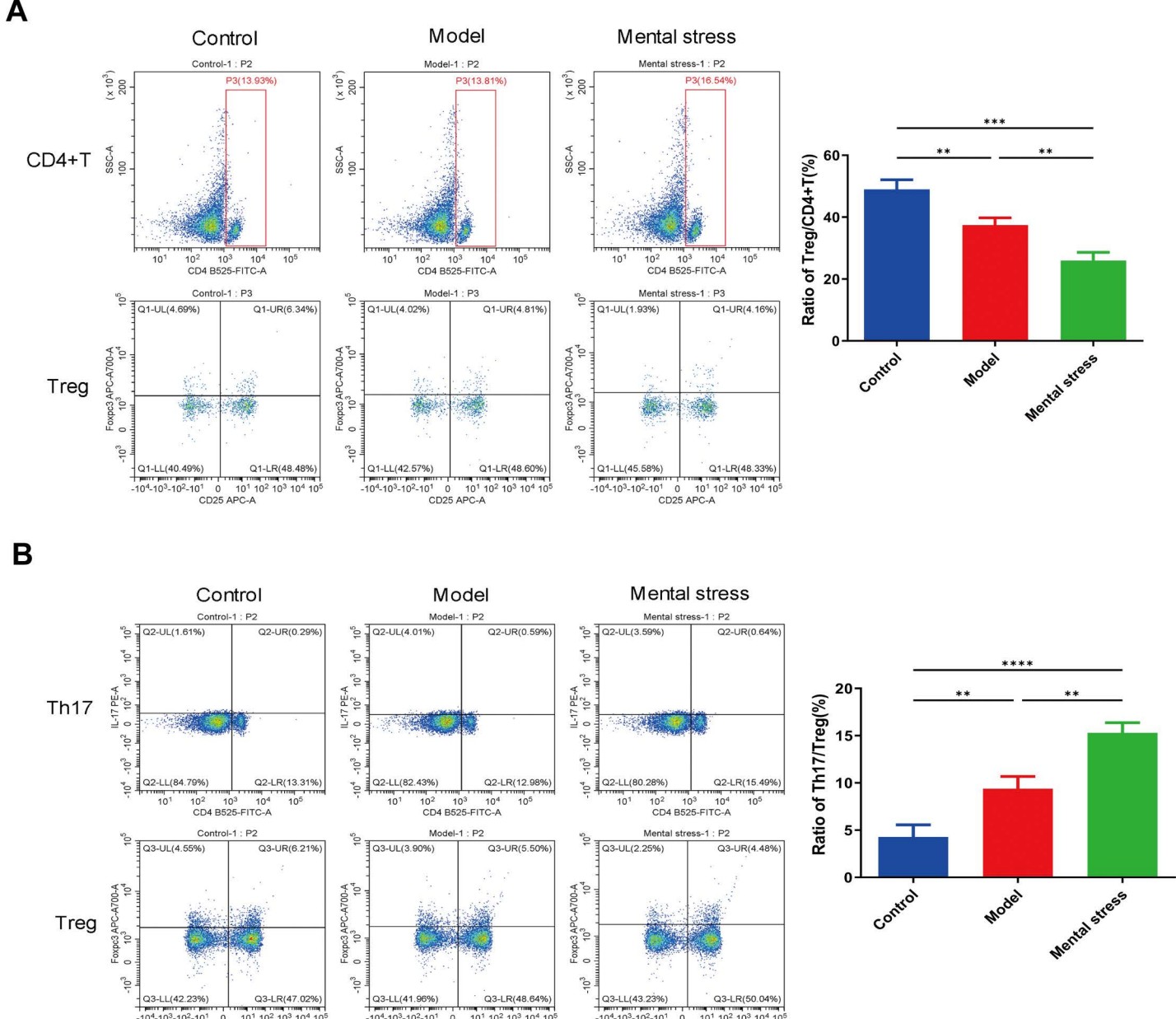

**Fig 3. Mental stress induces immune dysregulation in psoriatic mice. (A)**: Treg/CD4+T ratio in PBMCs of mice; **(B)**: Th17/Treg ratio in PBMCs of mice. $^{**}P < 0.01$, $^{***}P < 0.001$.

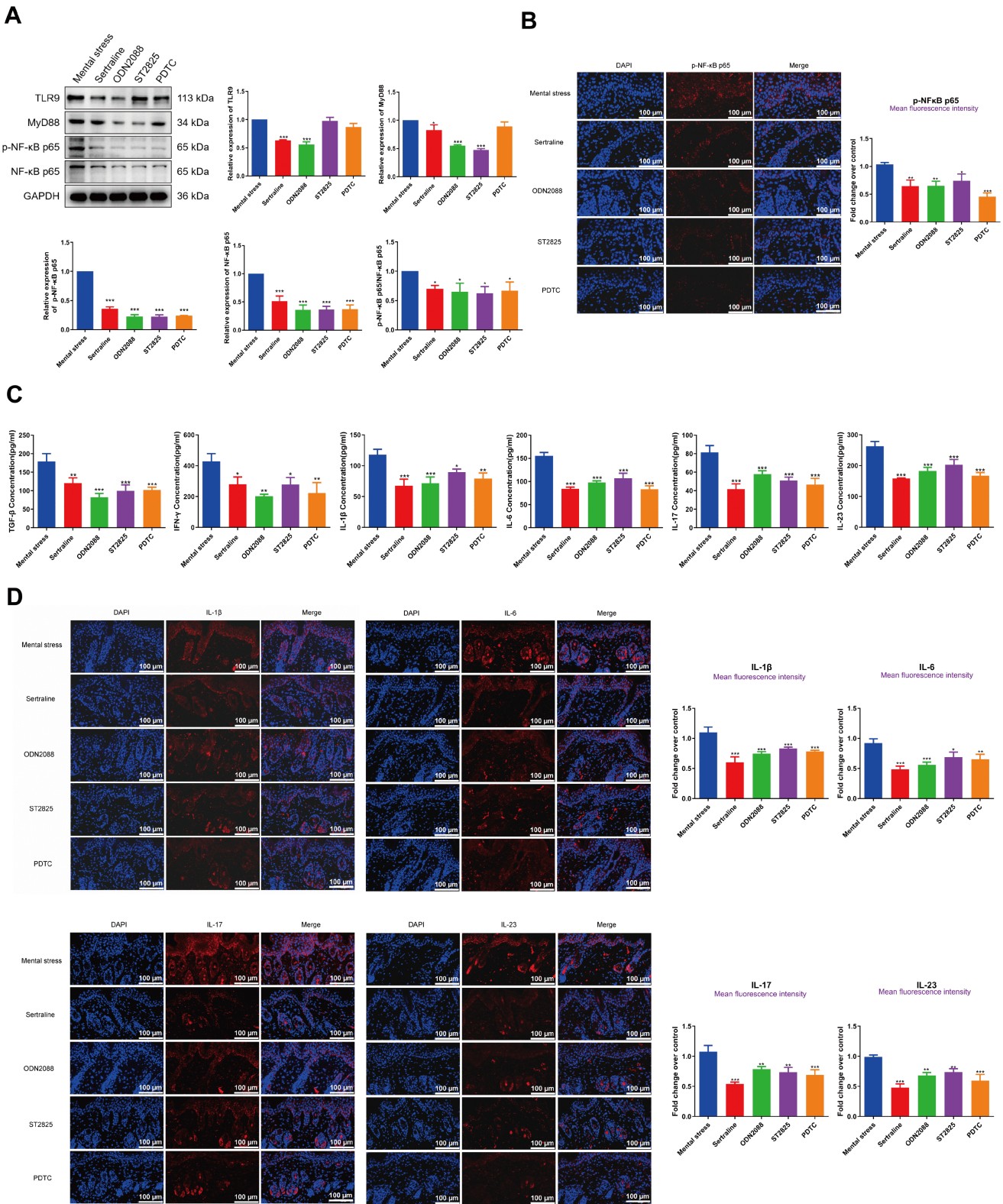

**Fig 4. Treatment with the inhibitors of the TLR9/MyD88/NF-κB pathway attenuates inflammation in mental stress-treated psoriatic mice.**
Following treatment with the specific inhibitors of the TLR9/MyD88/NF-κB pathway, (A): The protein levels of TLR9, MyD88, p-NF-κB p65 and NF-κB p65 in PBMCs of mice were measured by western blotting; (B) The fluorescence intensity of p-NF-κB p65 was assessed by immunofluorescence test, scale

bar = 100 μm; (C): The serum concentrations of TGF-β, IFN-γ, IL-1β, IL-6, IL-17 and IL-23 were detected by ELISA; (D) The fluorescence intensities of IL-1β, IL-6, IL-17 and IL-23 were assessed by immunofluorescence test, scale bar = 100 μm. *$P < 0.05$, **$P < 0.01$, ***$P < 0.001$.

the specific inhibitors of the TLR9/MyD88/NF-κB pathway suppressed the levels of inflammatory factors (Fig 4C-D). Interestingly, the administration of ODN2088, ST2825 or PDTC remarkably elevated the Treg/CD4+T ratio, and decreased the Th17/Treg ratio (Fig 5A-D).

### Knocking down the TLR9/MyD88 pathway alleviates skin lesions and inflammation in psoriatic mice

Next, we knocked down the expression of TLR9/MyD88 in mental stress-treated psoriatic mice. The PASI score showed that the skin lesions were markedly alleviated following the injection of LV-shTLR9 or LV-shMyD88 (Fig 6A), which was further verified HE staining (Fig 6B). Following injection of LV-shTLR9, we observed a significant decrease in

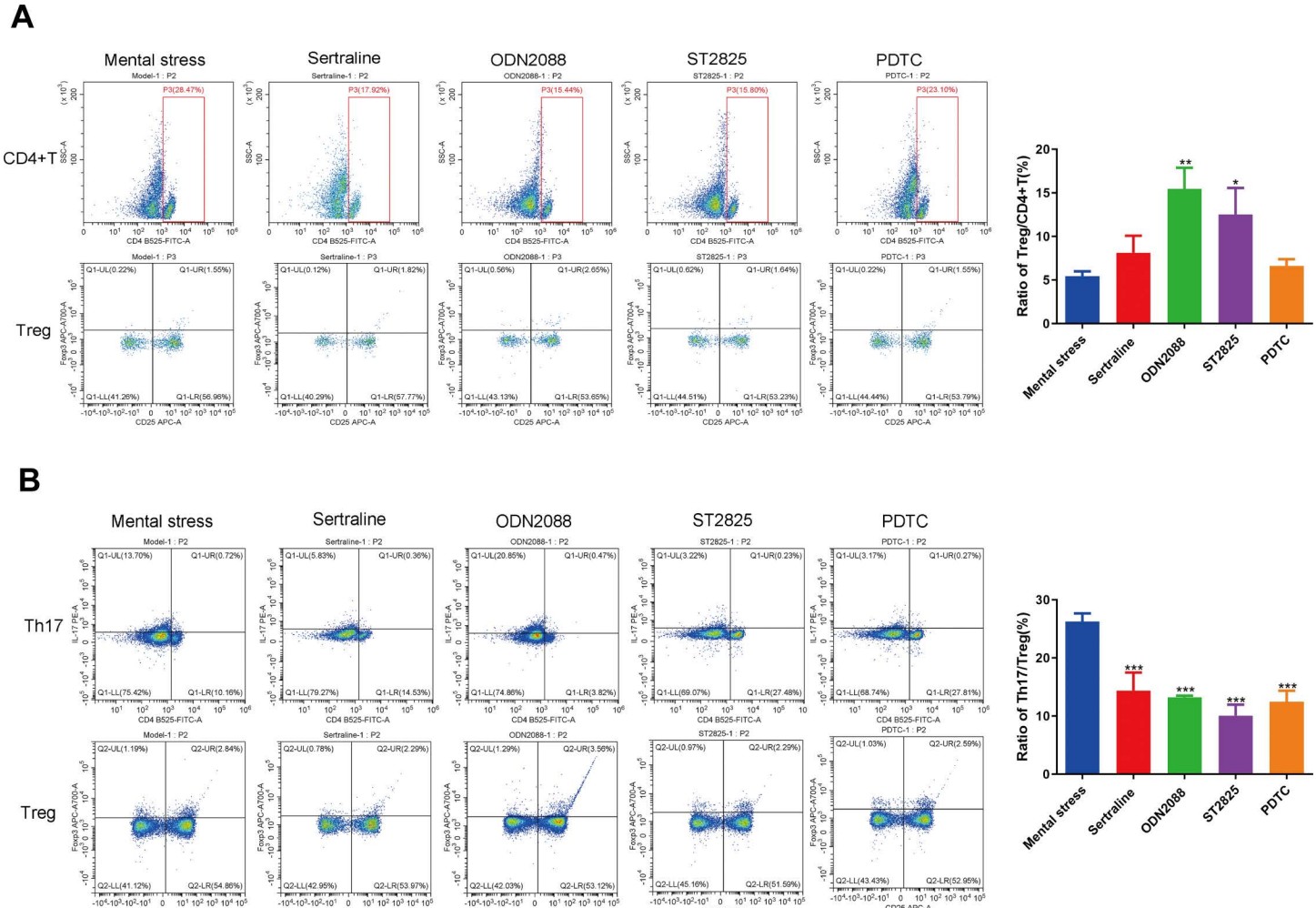

**Fig 5. Treatment with the inhibitors of the TLR9/MyD88/NF-κB pathway maintains immune balance in mental stress-treated psoriatic mice.** (A): Treg/CD4+ T ratio in PBMCs of mice; (B): Th17/Treg ratio in PBMCs of mice. *$P < 0.05$, **$P < 0.01$, ***$P < 0.001$.

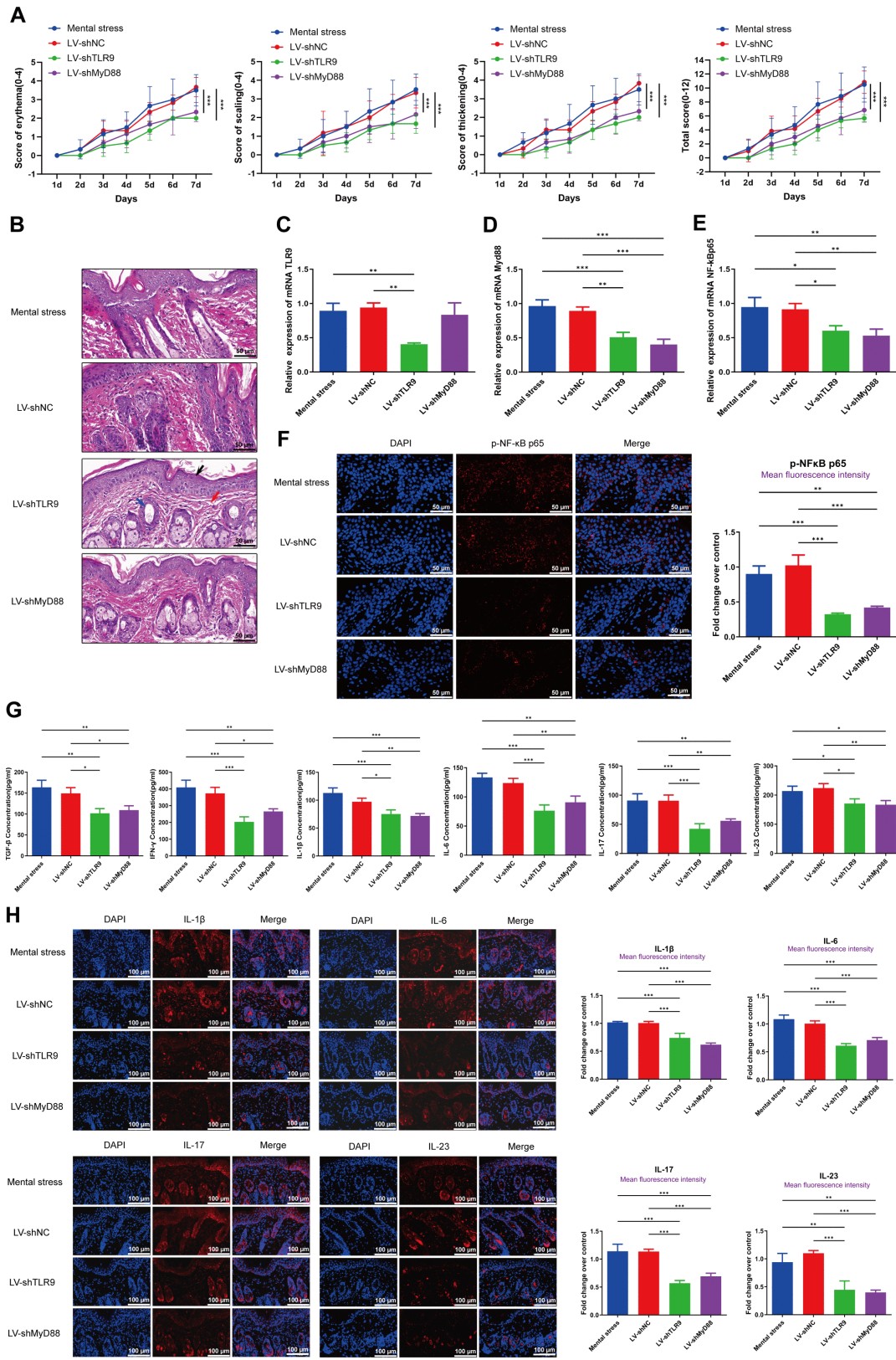

**Fig 6. Knocking down the TLR9/MyD88 pathway alleviates skin lesions and inflammation in psoriatic mice.** Following injection of LV-NC, LV-shTLR9, or LV-MyD88 in mice, **(A)**: Daily scores based on PASI. **(B)**: HE analysis of skin tissue in each group, scale bar = 50 μm; **(C-E)**: qRT-PCR

was used to detect the expression level of TLR9, MyD88, and NF-κB p65 mRNA in PBMCs of mice; **(F)** The fluorescence intensity of p-NF-κB p65 was assessed by immunofluorescence test, scale bar = 100 μm; **(G)**: The concentrations of TGF-β, IFN-γ, IL-1β, IL-6, IL-17 and IL-23 were detected by ELISA; **(H)** The fluorescence intensities of IL-1β, IL-6, IL-17 and IL-23 were assessed by immunofluorescence test, scale bar = 100 μm. $^*P < 0.05$, $^{**}P < 0.01$, $^{***}P < 0.001$.

TLR9 expression in skin lesions of mental stress-treated psoriatic mice (**Fig 6C**), while injection of both LV-shTLR9 and LV-shMyD88 markedly decreased the expression of MyD88, NF-κB p65, and p-NF-κB p65 (**Fig 6D-F**). Furthermore, the serum concentrations of TGF-β, INF-γ, IL-1β, IL-6, IL-17 and IL-23 in mental stress-treated psoriatic mice were also observed to be reduced upon the injection of LV-shTLR9 or LV-shMyD88 (**Fig 6G-H**).

## Discussion

Initially, psoriasis was considered a primary skin keratinization disorder. Around 1980, the association between immune dysregulation and psoriasis was revealed. With the advancement of the T cell hypothesis, psoriasis is now recognized as an immune disorder associated with to type 1 T cells [45]. Additionally, some researchers have indicated that most patients feel that mental stress exacerbates their condition and can recall stressful incidents before the onset or flare-up of psoriasis [17]. Additionally, individuals with psoriasis more commonly report remembering stressful experiences than those without the condition [17]. publicly available data suggest an association among mental stress, immune dysregulation, and psoriasis. In this study, we observed that mental stress stimulated inflammatory responses and aggravated skin lesions in an imiquimod-induced psoriatic mouse model. More importantly, the underlying mechanism was preliminarily elucidated, indicating that mental stress may mediate immune dysregulation through the activation of the TLR9/MyD88/NF-κB pathway, thereby promoting the progression of psoriasis.

The formation of inflammatory psoriatic plaques is caused by the infiltration of inflammatory cells into the skin [46]. Numerous inflammatory cytokines are aberrantly released during psoriasis progression. For example, IFN-γ interacts with endothelial cells and keratinocytes, resulting in their activation. It can promote the release of IL-1β, IL-6 and IL-23, which is a crucial biomarker for the disease activity of psoriasis [47]. Both IL-23 and IL-17 play key roles in keratinocyte hyperproliferation, thereby promoting the development of psoriasis [48]. In this study, mental stress was observed to significantly elevate the concentrations of inflammatory cytokines in psoriatic mice, including TGF-β, INF-γ, IL-1β, IL-6, IL-17 and IL-23. Additionally, skin lesions in psoriatic mice were further exacerbated under mental stress. These results suggested that mental stress promotes the occurrence and development of psoriasis. Apart from the aberrant release of inflammatory cytokines, the pathogenesis of psoriasis also involves in dysfunction of the immune system, particularly the CD4$^+$T cell subsets. Some reports have demonstrated that the number of IL-17-producing CD4$^+$T cell has been observed to be much higher in psoriatic skin lesions than in healthy skin [49]. Th17 cell activation plays a key role in the generation of IL-17, a cytokine that can either directly trigger or indirectly support inflammatory responses in keratinocytes [50]. Tregs are crucial for maintaining immune homeostasis and can lead to localized suppression of Th17 cells, thereby protecting against autoimmune disorders [51]. Therefore, preserving the delicate equilibrium between Th17 and Treg cells is essential for inhibiting psoriasis progression. In this study, following the induction of mental stress, the ratio of Treg/CD4$^+$T cells decreased, whereas the Th17/Treg ratio markedly increased in psoriatic mice. These results suggested that mental stress leads to immune dysregulation in mice, thereby promoting the development of psoriasis.

Previous studies have shown that the TLR9/MyD88/NF-κB pathway is strongly associated with inflammation-related diseases. For example, Jing et al. demonstrated that the TLR9/MyD88/NF-κB pathway aggravates inflammatory reactions during lung epithelial cell injury [52]. Li et al. reported that through inhibiting the activation of the TLR9/MyD88/NF-κB pathway, oxymatrine exhibits protective role in colitis [53]. Zhou et al. established an exercise-induced skeletal muscle

damage model in rat and reported that the TLR9/MyD88/NF-κB pathway is a potential target for acupuncture pretreatment on to attenuate inflammation [54]. However, the role of the TLR9/MyD88/NF-κB pathway in psoriasis progression remains under investigation. In this study, given that the expression of TLR, MyD88, and NF-κB p65 was increased in patients with psoriasis, and the induction of mental stress also activated this pathway in psoriatic mice, we speculated that the TLR9/MyD88/NF-κB pathway may be an underlying target of mental stress in stimulating inflammation during the progression of psoriasis. Our results showed that the inhibition of the TLR9/MyD88/NF-κB pathway significantly reduced the release of inflammatory cytokines in psoriatic mice under mental stress conditions, thereby verifying this hypothesis. More importantly, the development of psoriasis is driven by the aberrant expression of inflammatory cytokines and immune dysfunction. We further demonstrated that following administration of the inhibitors of the TLR9/MyD88/NF-κB pathway, the Treg/CD4$^+$T ratio was increased, whereasTh17/Treg ratio was decreased in mental stress-treated psoriatic mice. These results imply that inhibition of the TLR9/MyD88/NF-κB pathway is helpful for attenuating mental stress-mediated immune dysregulation in psoriatic mice.

## Conclusion

In conclusion, the TLR9/MyD88/NF-κB pathway serves as a potential target of mental stress mediated inflammatory responses and immune dysregulation. Mental stress may aggravate psoriasis development by activating this pathway. Our findings provide a preliminary insight into the interactions among mental stress, inflammation, immune dysregulation and the TLR9/MyD88/NF-κB pathway during the progression of psoriasis. The findings of in this study provide valuable information for developing potential clinical therapies targeting individuals with psoriasis.

## Supporting information

**S1 File. Raw images.**
(PDF)

## Author contributions

**Conceptualization:** Qipeng Xiao.

**Data curation:** Qipeng Xiao.

**Formal analysis:** Qiuhe Song, Dongyang Li, Zhihao Yuan, Chaowen Zhang, Jianqiao Wang, Fangfang Liao, Pengfei Xu.

**Funding acquisition:** Qiuhe Song, Dongyang Li.

**Investigation:** Qiuhe Song, Dongyang Li.

**Methodology:** Qiuhe Song, Dongyang Li, Zhihao Yuan, Chaowen Zhang, Jianqiao Wang, Fangfang Liao, Pengfei Xu.

**Project administration:** Qipeng Xiao.

**Resources:** Qipeng Xiao.

**Software:** Qiuhe Song, Dongyang Li, Zhihao Yuan, Chaowen Zhang, Jianqiao Wang, Fangfang Liao, Pengfei Xu.

**Supervision:** Qipeng Xiao.

**Validation:** Qipeng Xiao.

**Visualization:** Qiuhe Song, Dongyang Li, Zhihao Yuan, Chaowen Zhang, Jianqiao Wang, Fangfang Liao, Pengfei Xu.

**Writing – original draft:** Qiuhe Song, Dongyang Li.

**Writing – review & editing:** Qiuhe Song, Dongyang Li, Zhihao Yuan, Chaowen Zhang, Jianqiao Wang, Fangfang Liao, Pengfei Xu, Qipeng Xiao.

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
