## [Decision Letter · Decision Letter 0]

17 Nov 2025

Dear Dr. Xiao,

Thank you for submitting your manuscript to PLOS ONE. After careful consideration, we feel that it has merit but does not fully meet PLOS ONE’s publication criteria as it currently stands. Therefore, we invite you to submit a revised version of the manuscript that addresses the points raised during the review process.

We look forward to receiving your revised manuscript.

Kind regards,

Sadiq Umar

Academic Editor

PLOS ONE

Journal Requirements:

“This study was supported by National Natural Science Foundation of China (No. 82260283) and The Science and Technology Plan Project of Provincial Health Commission in 2025 (No. 202510823).”

5. We note that your Data Availability Statement is currently as follows: “All relevant data are within the manuscript and its Supporting Information files.”

Reviewers' comments:

Reviewer's Responses to Questions

**Comments to the Author**

1. Is the manuscript technically sound, and do the data support the conclusions?

Reviewer #1: Yes

Reviewer #2: Yes

2. Has the statistical analysis been performed appropriately and rigorously?

Reviewer #1: Yes

Reviewer #2: Yes

3. Have the authors made all data underlying the findings in their manuscript fully available?

Reviewer #1: Yes

Reviewer #2: Yes

4. Is the manuscript presented in an intelligible fashion and written in standard English?

Reviewer #1: Yes

Reviewer #2: Yes

Reviewer #1: In the manuscript titled “TLR9/MyD88/NF-κB signaling mediates mental stress-induced exacerbation of psoriasis through immune dysregulation in a mouse model”, this article mainly explores the mechanisms by which mental stress exacerbates human psoriasis. The results first showed that TLR9/MyD88/NF-κB pathway related molecules were significantly upregulated in peripheral blood mononuclear cells of psoriasis patients and psoriasis mouse models, while the expression of inflammatory factors was elevated. Subsequently, the article found for the first time that mental stress was shown to activate the TLR9/MyD88/NF - κ B pathway, enhance the production of inflammatory factors, increase the Th17/Treg ratio, and decrease the Treg/CD4+T ratio. The use of antipsychotic drugs or pathway inhibitors can inhibit this change. Overall, these findings provide valuable insights into the pathophysiological mechanisms underlying the progression of psoriasis, particularly elucidating the immune regulatory axis mediated by mental stress. But there are the following key issues that affect its publication value:

1. What is the relationship between TLR9/NF-κB pathway and Th17/Treg ratio, as well as Treg/CD4+T ratio? Explain in the introduction why these two ratios were chosen as detection indicators to ensure contextual coherence

2. Phosphorylated NF-κB P65 also plays a very important role in the inflammatory process, especially in this inflammatory axis. Why not study p-NF-κB P65?

3. The changes in NF-κB protein are generally not simply an increase or decrease in quantity. The process of inflammatory activation is accompanied by complex nuclear translocation and other processes. Why has no further exploration been conducted? Please provide a detailed explanation

4. The changes in the levels of inflammatory axis proteins and mRNA are very significant, and if validated from the perspective of cellular localization, it can further increase credibility. For example, immunofluorescence

5. Line151 Please provide references for Internationally recognized method of solitary stress

6. Interpret the results of the flow cytometry and how to determine the changes in Th17/Treg and Treg/CD4+T ratios based on the images of the flow cytometry results

7. Although the results of HE staining are obvious, I believe that adding a comparison of gross photographs of mouse skin would be more intuitive and persuasive.

8. The current manuscript is recommended to be polished by professional language editing service personnel.

Summary

This article needs to further improve the overall experimental logic and specific details.

Reviewer #2: This study investigates the mechanism by which psychological stress exacerbates psoriasis via the TLR9/MyD88/NF-κB signaling pathway. Utilizing both patient samples and a mouse model, the authors demonstrate that mental stress activates this pathway, promotes the release of inflammatory factors, and induces an imbalance in the Th17/Treg cell ratio. Inhibition of this pathway using pharmacological inhibitors and gene knockdown techniques alleviated the inflammatory response, improved immune balance, and mitigated skin lesions. The research reveals a novel mechanism through which mental stress aggravates psoriasis via immune dysregulation, offering a potential therapeutic target. The manuscript is recommended for publication after minor revisions.

Specific points for consideration:

1. In Figure 1, the data appear to show considerable variability among replicates. Furthermore, the selection of the Y-axis scale in panel G seems inappropriate and may misrepresent the data.

2. For the histological images presented in Figure 2A and Figure 6B, it is essential to include scale bars to provide context for the magnification.

3. The description of the "single-housing stress" model should be elaborated. Providing behavioral data validating the effectiveness of this stress paradigm is highly recommended.

4. Certain sections of the Materials and Methods could be condensed for brevity and clarity.

5. In Figure 4A, the bar representing the data for the control group appears to be missing. Please clarify or correct this representation.

6. For the H&E staining images, using boxes or arrows to highlight specific pathological features would greatly enhance clarity for the reader.

7. For references on inflammation and immune system discussions,PMID: 35032680, PMID: 38453637, and PMID: 37573970.

**Do you want your identity to be public for this peer review?** For information about this choice, including consent withdrawal, please see our Privacy Policy

Reviewer #1: **Yes:** Chengye Che

Reviewer #2: No

---

## [Author Response · Author response to Decision Letter 1]

4 Jan 2026

Responds to editor

Response: Thanks for your advice. We have revised the manuscript and ensured that it meets PLOS ONE's style requirements.

“This study was supported by National Natural Science Foundation of China (No. 82260283) and The Science and Technology Plan Project of Provincial Health Commission in 2025 (No. 202510823).”

Response: Thanks for your advice. We feel sorry for our carelessness. In fact, the Acknowledgments Section should state “Not applicable”, while the statement “This study was supported by National Natural Science Foundation of China (No. 82260283) and The Science and Technology Plan Project of Provincial Health Commission in 2025 (No. 202510823)” should appear in the Funding Statement Section.

According to the requirements, we have amended the statements within cover letter.

Response: Thanks for your advice. As mentioned in comment 2, we have amended the statements within cover letter.

Response: Thanks for your advice. As mentioned in comment 2, we have amended the statements within cover letter.

5. We note that your Data Availability Statement is currently as follows: “All relevant data are within the manuscript and its Supporting Information files.”

Response: Thanks for your advice. All raw data of this study have been uploaded in Zenodo. Additionally, we have revised the Data Availability Statement as “The datasets generated and/or analyzed during the current study are available in Zenodo, the open-access repository operated by CERN, [https://zenodo.org/records/18069564?token=eyJhbGciOiJIUzUxMiJ9.eyJpZCI6ImZjYzdkNmNlLTFhZjktNDE3ZS05YzdiLTI2NjhmM2U4MWY1OSIsImRhdGEiOnt9LCJyYW5kb20iOiJmYTBjMmIyZDAyNjk2Nzg0NTgxNGJjODRhMTExNDFkZCJ9.HeVpzwu5n4OdiSw3toFMjTZBCfdZc_uQMNAjmFJ4yxSigNemmWOl5bSr0zyMPHrk74bYpXBmdUajlgSLbbN7-w]”

Response: Thanks for your advice. The original uncropped and unadjusted gels of this study have been uploaded in Zenodo, the open-access repository operated by CERN, [https://zenodo.org/records/18069564?token=eyJhbGciOiJIUzUxMiJ9.eyJpZCI6ImZjYzdkNmNlLTFhZjktNDE3ZS05YzdiLTI2NjhmM2U4MWY1OSIsImRhdGEiOnt9LCJyYW5kb20iOiJmYTBjMmIyZDAyNjk2Nzg0NTgxNGJjODRhMTExNDFkZCJ9.HeVpzwu5n4OdiSw3toFMjTZBCfdZc_uQMNAjmFJ4yxSigNemmWOl5bSr0zyMPHrk74bYpXBmdUajlgSLbbN7-w]. According to the requirements, we have clarified this in cover letter.

Response: Thanks for your advice. As you mentioned above, the comment 7 of reviewer 2 include a recommendation to cite three previously published works (PMID: 35032680, PMID: 38453637, and PMID: 37573970). After a careful review of these three references, we summarize their core themes as follows:

-PMID: 35032680: Lycopene attenuates oxidative stress, inflammation, and apoptosis by modulating Nrf2/NF-κB balance in sulfamethoxazole-induced neurotoxicity in grass carp (Ctenopharyngodon Idella)

This study investigates the protective effect of lycopene against sulfamethoxazole-induced neurotoxicity in grass carp (Ctenopharyngodon idella), demonstrating that lycopene effectively alleviates oxidative stress, inflammation and apoptosis in the fish’s nervous system by regulating the balance of the Nrf2 and NF-κB signaling pathways, thus providing a potential strategy to mitigate the adverse impacts of sulfamethoxazole exposure in aquaculture.

-PMID: 38453637: Endoplasmic reticulum stress-induced NLRP3 inflammasome activation as a novel mechanism of polystyrene microplastics (PS-MPs)-induced pulmonary inflammation in chickens

This study identifies that endoplasmic reticulum stress-triggered NLRP3 inflammasome activation serves as a novel mechanism underlying polystyrene microplastics (PS-MPs)-induced pulmonary inflammation in chickens, revealing that PS-MPs exposure causes pulmonary pathological damage, initiates endoplasmic reticulum stress, and further activates NLRP3 inflammasome to induce pyroptosis and inflammatory responses via relevant signaling pathways including NF-κB, thus providing new insights into the pulmonary toxicological mechanisms of microplastics.

- PMID: 37573970: Evodiamine alleviates DEHP-induced hepatocyte pyroptosis, necroptosis and immunosuppression in grass carp through ROS-regulated TLR4 / MyD88 / NF-κB pathway

This study investigates the protective effect of evodiamine against DEHP-induced hepatotoxicity in grass carp, demonstrating that DEHP activates the ROS-regulated TLR4/MyD88/NF-κB pathway to trigger hepatocyte pyroptosis, necroptosis, and immunosuppression, while evodiamine alleviates these adverse effects by inhibiting ROS production, blocking pathway activation, and restoring hepatocellular immune function.

These three studies are not only irrelevant to our research in terms of disease types but also exhibit significant differences in species (mouse, fish, chicken), target organs (skin, liver, lung, nerve), and initiating stimuli (mental stress, pollutants, drugs). Therefore, after careful evaluation, we decided not to cite these three references in accordance with the editor’s requirements.

Responds to the reviewers

Reviewer Comments

1. Is the manuscript technically sound, and do the data support the conclusions?

Reviewer #1: Yes

Reviewer #2: Yes

Response: Thanks for your approval.

2. Has the statistical analysis been performed appropriately and rigorously?

Reviewer #1: Yes

Reviewer #2: Yes

Response: Thanks for your approval.

3. Have the authors made all data underlying the findings in their manuscript fully available?

Reviewer #1: Yes

Reviewer #2: Yes

Response: Thanks for your approval.

4. Is the manuscript presented in an intelligible fashion and written in standard English?

Reviewer #1: Yes

Reviewer #2: Yes

Response: Thanks for your approval.

5. Review Comments to the Author

Reviewer #1:

In the manuscript titled “TLR9/MyD88/NF-κB signaling mediates mental stress-induced exacerbation of psoriasis through immune dysregulation in a mouse model”, this article mainly explores the mechanisms by which mental stress exacerbates human psoriasis. The results first showed that TLR9/MyD88/NF-κB pathway related molecules were significantly upregulated in peripheral blood mononuclear cells of psoriasis patients and psoriasis mouse models, while the expression of inflammatory factors was elevated. Subsequently, the article found for the first time that mental stress was shown to activate the TLR9/MyD88/NF - κ B pathway, enhance the production of inflammatory factors, increase the Th17/Treg ratio, and decrease the Treg/CD4+T ratio. The use of antipsychotic drugs or pathway inhibitors can inhibit this change. Overall, these findings provide valuable insights into the pathophysiological mechanisms underlying the progression of psoriasis, particularly elucidating the immune regulatory axis mediated by mental stress. But there are the following key issues that affect its publication value:

1. What is the relationship between TLR9/NF-κB pathway and Th17/Treg ratio, as well as Treg/CD4+T ratio? Explain in the introduction why these two ratios were chosen as detection indicators to ensure contextual coherence.

Response: Thanks for your advice. According to the requirements, we have explained the queries mentioned in the above comments. The detailed descriptions are in the second and third paragraph of Introduction section and shown as follows (red):

Mental stress plays a key role in the development of several skin diseases. The stigma associated with these conditions may exacerbate psychological burden, thereby creating a vicious cycle that further contributes to the development of skin diseases (18). The prevalence of psoriasis is notably higher among individuals who experience major stressful events (19). Furthermore, mental stress is the second most important factor contributing to psoriasis exacerbation after seasonal changes (20). Recent studies have suggested that mental stress modulates skin immune imbalance to promote psoriasis progression, which is associated with the Toll-like receptor (TLR) pathway (18, 21), and regulates the expression of cytokines such as IL-17 and IL-23 (22-26). Th17 cells exacerbate skin inflammatory infiltration by secreting pro-inflammatory cytokines such as IL-17 and IL-23, whereas regulatory T (Treg) cells maintain immune homeostasis through the secretion of anti-inflammatory cytokines including IL-10 and transforming growth factor-β (TGF-β). The dynamic balance between these two cell subsets, reflected by the Th17/Treg ratio, directly regulates the inflammatory severity and disease progression of psoriasis [PMID: 36444619]. Clinical studies have confirmed that the Th17/Treg ratio is significantly elevated in the peripheral blood and lesional tissues of psoriasis patients [PMID: 37596509]. Therefore, this ratio is regarded as a core indicator for evaluating immune dysregulation in psoriasis. Meanwhile, as a critical subset of CD4⁺T cells, Treg/CD4⁺T ratio

---

## [Decision Letter · Decision Letter 1]

23 Feb 2026

TLR9/MyD88/NF-κB signaling mediates mental stress-induced exacerbation of psoriasis through immune dysregulation in a mouse model

PONE-D-25-36503R1

Dear Dr. Xiao,

We’re pleased to inform you that your manuscript has been judged scientifically suitable for publication and will be formally accepted for publication once it meets all outstanding technical requirements.

Kind regards,

Sadiq Umar

Academic Editor

PLOS One

Additional Editor Comments (optional):

Recommend for acceptance.

---

## [Editor Report · Acceptance letter]

PONE-D-25-36503R1

PLOS One

Dear Dr. Xiao,

I'm pleased to inform you that your manuscript has been deemed suitable for publication in PLOS One. Congratulations! Your manuscript is now being handed over to our production team.

Kind regards,

on behalf of

Dr. Sadiq Umar

Academic Editor

PLOS One